# Postgraduate Research Flow Experience Scale: Development and Preliminary Validation

**DOI:** 10.3390/bs15111453

**Published:** 2025-10-25

**Authors:** Fang Liu, Ruofei Du, Qian Lu, Yulu Liu, Jin Wang

**Affiliations:** 1Key Research Base of Humanities and Social Sciences of the Ministry of Education, Academy of Psychology and Behavior, Tianjin Normal University, Tianjin 300387, China; wangjin@tjnu.edu.cn; 2Faculty of Psychology, Tianjin Normal University, Tianjin 300387, China; 3Luoning No. 1 Senior High School, Luoyang 471700, China; 4Tianjin Jinghai Affiliated School of Beijing Normal University, Tianjin 301600, China

**Keywords:** postgraduate, academic research, flow experience, scale development

## Abstract

Background: This study developed the Postgraduate Research Flow Experience Scale and examined its psychometric properties, including reliability and validity. Methods: The development of a preliminary scale draft was guided by initial interviews with 42 participants. Item analysis and exploratory factor analysis (EFA) were subsequently conducted on data collected from 133 participants. Finally, psychometric evaluations, including reliability and validity tests, were performed using data from 944 participants. Results: The final version of the Postgraduate Research Flow Experience Scale consists of 15 items distributed across three dimensions: Clear Goals, Concentration, and Autotelic Experience. This scale demonstrated satisfactory psychometric properties, including construct validity, criterion validity, internal consistency reliability, split-half reliability, and composite reliability, meeting established psychometric standards. Conclusion: The Postgraduate Research Flow Experience Scale developed in this study demonstrates robust psychometric properties, establishing its suitability as a measurement tool for assessing flow experience in postgraduate scientific research.

## 1. Introduction

Flow, originally conceptualized by Csikszentmihalyi, denotes an optimal psychological state characterized by intense focus, deep immersion, and intrinsic enjoyment during activity. This state emerges when perceived challenges are balanced by an individual’s skills ([5], [6]) and yields multifaceted benefits, including heightened happiness ([9]), enhanced creativity ([19]), as well as improved talent development and self-esteem ([2], [3]).

Systematic analyses identify nine core characteristics of flow: clear goals, immediate feedback, balance between challenge and skill, intense concentration, merging of action and awareness, sense of control, loss of self-consciousness, transformation of time, and autotelic experience ([4]; [8]). Researchers further organize these characteristics into three sequential stages: flow antecedents, flow experience, and flow consequences ([3]). Flow antecedents refer to factors triggering flow, such as clear goals and immediate feedback. Flow experience denotes the core state manifested during flow, characterized by intense concentration and time distortion. Flow consequences encompass emotional and behavioral changes resulting from flow, including purposeful experience ([10]; [14]).

Flow can arise in virtually any activity ([22]). In learning contexts, flow enhances engagement ([1]), fosters intrinsic interest, and improves performance ([23]). High-flow experiences strengthen intrinsic motivation, encouraging learners to re-engage and further improve through deliberate practice ([16]). Flow also enhances self-esteem, academic achievement, and satisfaction ([15]). Moreover, flow reduces anxiety, stress, and boredom while promoting self-actualization ([21]; [24]). Research indicates that low-flow states correlate with high levels of academic procrastination, conversely underscoring flow’s positive impact on learning ([17]). Building on existing evidence, flow experience in learning demonstrates significant pedagogical influence, particularly regarding cognitive engagement and motivational outcomes.

During postgraduate education, research activities constitute a specialized form of learning. Through participation in research projects, postgraduate students acquire knowledge, enhance competencies, and achieve professional development. This process constitutes a high-order cognitive practice characterized by three distinguishing features. First, the autonomous nature of research demands sustained metacognitive awareness. Effective knowledge construction and competency development occur only through focused allocation of attentional resources. Second, the nonlinear trajectory of knowledge innovation generates phased fluctuations. Here, clearly defined sub-goals serve as pivotal anchors for sustaining progress. Third, enduring intrinsic interest in research generates persistent impetus—the core driver of sustainable research competence development. Consequently, experiencing flow in scientific research is crucial for fostering sustained engagement and intrinsic interest; its absence may contribute to the psychological distress and pressures commonly encountered during postgraduate studies.

As a central concept in positive psychology, flow has prompted ongoing debate regarding the contextual adaptability of its measurement tools. Classical instruments such as the Flow State Scale-2 (FSS-2) ([13]) have shown limitations in educational settings, where certain dimensions are difficult to capture via self-report questionnaires. The nine dimensions of the FSS-2 can be clearly distinguished in contexts such as sports and gaming, while its factor structure proves unstable in educational task scenarios, failing to reliably extract all nine factors ([12]). Some items in the FSS-2 are phrased too generically and cannot accurately capture the flow experience specific to educational or research activities. Scientific research endeavors typically require sustained commitment over extended periods, and the challenges inherent in such work often fluctuate significantly due to external factors such as negative results, peer review feedback, technical issues, or equipment failures. The core challenge stems from the dynamic context-dependency of flow characteristics. When individuals engage in activities with distinct attributes, context-specific factors attenuate the salience of certain experiential dimensions in conscious awareness. This attenuation consequently undermines the capacity of traditional measurement instruments to comprehensively assess these manifestations.

Classical instruments, such as the FSS-2, focus on the characteristics of the present-moment experience and therefore do not capture the full dynamic of flow. Graduate research typically unfolds in multiple stages—literature review, protocol design, experiment execution, data collection, data analysis, paper writing, submission, and revision. Examining research activities requires looking at the situational variables present at task initiation, the subjective experiences that unfold during the task, and the behavioral and psychological outcomes that follow its completion. Existing questionnaires are inadequate for measuring the unique cognitive immersion and evolving flow experiences within research tasks. Building on the foregoing, the present study integrates the unique features of postgraduate research contexts and systematically constructs a measurement model of research-related flow grounded in the three-stage model of flow ([3]; [10]; [20]). The three-stage model of flow, with its “flow antecedents, flow experience, and flow consequences” framework, can fully capture the dynamic process of flow and thus aligns well with the characteristics of graduate research activities. To address the contextual limitations of existing scales, we developed a purpose-built instrument that precisely assesses the flow experiences of postgraduates engaged in research training. The scale development serves four key purposes: first, to deepen understanding of flow dynamics within scientific research contexts; second, to provide researchers with a metacognitive tool to identify optimal experiential patterns; third, to establish a framework enabling supervisors to deliver targeted guidance that fosters research capacity; and fourth, to create context-specific intervention targets for institutional mental-health support systems, thereby promoting psychological resilience in academic communities.

## 2. Methods and Results

### 2.1. Open-Ended Questionnaire Survey

#### 2.1.1. Participants

Forty-two postgraduate students participated in semi-structured interviews (24 male, 18 female; Mean age = 23.98 ± 1.83). Disciplines: Education (19%), Economics/Management (11.9%), Science/Engineering (45.2%), Humanities/History (14.3%), Other (9.5%). Degree types: PhD student (19%), Academic Master’s student (31%), Professional Master’s student (50%).

#### 2.1.2. Initial Item Generation

Interviews covered topics like the nature of research in their field, typical activities, publication processes, and personal research experiences. Based on interview content analysis and the three stages model of flow, 29 initial items were generated (9 Antecedents, 10 Experience, 7 Consequences). After review by psychology PhDs and Masters, redundant, unclear, or ambiguous items were revised or removed, resulting in a 26-item preliminary questionnaire. Responses were recorded on a 5-point Likert scale (1 = “Strongly Disagree”, 5 = “Strongly Agree”). Dimension scores were summed; higher scores indicated greater research flow experience.

### 2.2. Preliminary Testing of the Scale

#### 2.2.1. Participants

An online survey recruited 142 postgraduates; 133 valid responses remained after data cleaning (40 male, 93 female; Mean age = 23.98 ± 1.90). Disciplines: Education (33.8%), Economics/Management (20.3%), Science/Engineering (21.8%), Humanities/History (18%), Other (6%). Degree types: PhD student (6.1%), Academic Master’s student (41.2%), Professional Master’s student (51.9%), Other (0.8%).

#### 2.2.2. Item Analysis

Participants were divided into high- (top 27%) and low- (bottom 27%) scoring groups based on total scores. Independent samples t-tests revealed one item with non-significant differences between groups, and this item was deleted, leaving 25 items.

#### 2.2.3. Difficulty Analysis (Popularity Analysis)

Difficulty was assessed using popularity analysis for non-dichotomous items. All items met criteria for multivariate normality (skewness < |3|, kurtosis < 10). Popularity indices (*p* = Mean score / Max possible score) ranged from 0.62 to 0.80 for all 25 items, indicating acceptable difficulty; all items were retained.

#### 2.2.4. Exploratory Factor Analysis (EFA)

Data suitability for EFA was confirmed (KMO = 0.90, Bartlett’s test *p* < 0.001). Principal component analysis with varimax rotation was performed on the 25 items. Items were deleted based on: factor loading < 0.40, cross-loading (difference < 0.20 between highest and next highest loading), or factors with fewer than 3 items. Iterative analysis yielded a final 3 factor solution with 15 items, explaining 65.671% of the total variance (Table 1).

The three factors were named sequentially based on the content reflected by each item. Factor 1, composed of 6 items, primarily reflects postgraduates’ sensitivity to time and high concentration during research activities, which corresponds to characteristics of the flow experience phase. It was named “Concentration” and accounts for 26.218% of the explained variance. Factor 2, comprising 5 items, mainly relates to goal setting before research activities. The experience of flow requires clear goal establishment, which belongs to the antecedent phase of flow. This factor was named “Clear Goals” and explains 20.758% of the variance. Factor 3, consisting of 4 items, focuses on the sense of happiness derived from research activities. This happiness stems from individuals’ intrinsic interest in research itself, which sustainably motivates them to re-engage in research activities through internal drive. It reflects feelings during the outcome phase of flow and was named “Conscious Experience,” explaining 18.695% of the variance.

### 2.3. Formal Testing of the Scale

#### 2.3.1. Participants

An online survey recruited 1168 postgraduates; 944 valid responses remained after data cleaning (590 male, 354 female; Mean age = 24.29 ± 2.42). Disciplines: Education (11.5%), Economics/Management (21.2%), Science/Engineering (48.3%), Humanities/History (15.6%), Other (3.4%). Degree types: PhD student (27%), Academic Master’s student (30.5%), Professional Master’s student (40.1%), Other (2.4%).

#### 2.3.2. Validity Analysis

##### Construct Validity

Construct validity was examined via Confirmatory Factor Analysis (CFA). The hypothesized three-factor model of the Postgraduate Research Flow Experience Scale was estimated with Mplus 8.0. Model fit indices (Table 2) and satisfied conventional psychometric benchmarks. Standardized factor loadings ranged from 0.63 to 0.76 (Figure 1).

##### Criterion Validity

Given the established link between learning flow and research flow, the EduFlow Scale ([11]) measures four flow dimensions: Cognitive Absorption, Transformation of Time, Loss of Self-consciousness, and Autotelic Experience. Well-being was used as the criterion. The results showed that the overall score and its three subscales correlated significantly and positively with the EduFlow Scale (*p* < 0.01) (Table 3), supporting the scale’s good criterion validity.

#### 2.3.3. Reliability Analysis

Reliability analyses (Table 4) indicated strong internal consistency for the Postgraduate Research Flow Experience Scale. Cronbach’s α was 0.913 for the full scale and ranged from 0.818 to 0.841 across the three subscales. Guttman split-half reliability was 0.826 for the total score and 0.786–0.814 for the subscales. McDonald’s omega was 0.914 for the overall scale and 0.820–0.843 for each subscale. These values meet conventional psychometric thresholds, attesting to the instrument’s reliability.

## 3. Discussion

### 3.1. Reliability and Validity of the Postgraduate Research Flow Experience Scale

This study developed the Postgraduate Research Flow Experience Scale based on the characteristics of postgraduate research activities and the three-stage model of flow. Reliability analyses showed that internal-consistency, split-half, and composite reliability estimates for both the overall scale and its three subscales were satisfactory and met psychometric standards, indicating strong internal consistency and stability. Confirmatory factor analysis (CFA) yielded good model fit, and criterion-related validity analyses revealed a significant, high correlation with the EduFlow Scale, confirming robust criterion validity.

### 3.2. Structure of the Postgraduate Research Flow Experience Scale

Empirical evaluation confirmed that the derived dimensions align closely with the theoretical framework. The researchers named the three extracted factors based on the content reflected in the items and the three-stage model of flow. Postgraduate research flow unfolds across three sequential stages: Clear Goals (Flow antecedents), Concentration (Flow experience), and Autotelic Experience (Flow consequences).

#### 3.2.1. Clear Goals: The Triggering Prerequisite for Research Flow Experience

Clear goals refer to postgraduates’ explicit and stable understanding of specific outcomes, quantitative criteria, and evaluation standards required for short-term or long-term research tasks, thereby providing directional reference for subsequent research activities. Clear and challenging goals are a prerequisite for inducing flow ([4]). Postgraduates need to establish explicit, stage-specific goals throughout their research. At the macro-level, they need to define the overarching direction of their research agenda. At the meso-level, they should formulate phased research plans. At the micro-level, they must break these goals into specific tasks such as reading literature and mastering methodology. Achieving self-set goals not only fosters a sense of accomplishment and motivates research engagement but also enhances research capability, enabling effective resolution of academic challenges and improved efficiency ([25]).

#### 3.2.2. Concentration: The Core Feature of Research Flow Experience

Concentration denotes a psychological state in which individuals highly focus attentional resources on current task stimuli during research activities, suppressing external distractions and internal irrelevant thoughts to achieve optimal work efficiency. Research indicates that when individuals are fully absorbed in what they are doing, the flow experience is both pleasurable and invigorating ([7]). When a person enters a self-transcendent state of deep focus, the intense yet orderly allocation of attention optimizes cognitive resources. In research activities, such concentration inevitably boosts efficiency. Studies confirm that research engagement is critical for fostering concentration; sustained engagement allows postgraduates to immerse themselves and perceive the value of their work ([18]). This deep immersion not only improves task execution but may also stimulate creative thinking via neuroplastic mechanisms.

#### 3.2.3. Autotelic Experience: The Sustained Motivation for Research Flow Experience

Autotelic experience, which is defined as the intrinsic well-being derived from the activity itself and is characterized by spontaneous feelings of pleasure, satisfaction, and self-efficacy generated during the execution of research tasks, reflects the flow consequences stage. This well-being stems from genuine interest in the activity, which continually drives re-engagement and fosters autonomous motivation. Autotelic experience describes a phenomenon where scientific research itself becomes an intrinsic reward source. Unconditioned by external rewards, this pure enjoyment is positively reinforced through the dopaminergic reward system, creating a positive feedback loop for research behavior. Postgraduates with heightened autotelic experience develop deeper interest in research and stronger intrinsic motivation. Moreover, higher intrinsic motivation is closely associated with greater self-efficacy. Previous research has demonstrated that postgraduates with high self-efficacy are more confident in completing tasks; when faced with difficulties, they frame obstacles as opportunities for innovation, thereby sustaining robust intrinsic motivation ([26]). Therefore, autotelic experience functions as a form of psychological capital accumulation, ultimately translating into sustained willingness to engage in research and generating a virtuous “efficacy–motivation–engagement” cycle.

Developed based on the three stages model of flow experience, this scale could not only diagnose postgraduates’ current research flow states but could also predict their subsequent persistence and re-engagement intentions in research activities, demonstrating significant practical utility. In the future, personalized training interventions can be delivered to students based on their dimensional scores on the Postgraduate Research Flow Experience Scale. For students with low concentration or unclear goals, phased goal-setting training can be provided to help them practice time management techniques. For those scoring low on the “Autotelic Experience” dimension, research milestones can be broken down with visual progress trackers, assisting graduate students in finding their optimal pace during the extended research journey. Furthermore, this scale is suitable for large-scale administration. When a postgraduate student’s scores fall below critical thresholds, the system can automatically alert supervisors or psychological service centers to enable early intervention, thereby preventing research fatigue from escalating into more severe consequences such as depression or withdrawal. Additionally, the instrument provides quantifiable process indicators for mentoring quality and training outcomes to supervisors, academic departments, and educational authorities.

This scale is designed for postgraduate students actively engaged in research activities and should be administered in relatively stable research settings. It is essential to ensure that participants possess normal cognitive comprehension abilities. Research flow experiences are susceptible to various influencing factors. First, attention should be paid to participants’ intrinsic factors during scale administration, including emotional fluctuations (e.g., anxiety and stress may significantly reduce concentration scores), motivation levels (lack of intrinsic motivation may distort autotelic experience scores), and personal emergencies (e.g., health issues may temporarily affect overall flow states). Second, external interfering factors should also be considered, such as environmental elements (laboratory noise, spatial crowding that may disrupt cognitive focus), technical failures (data loss, software crashes that could interrupt research activities), and equipment deficiencies (underperforming computers that may delay task progress). When interpreting scale results, the potential interactive effects of these intrinsic and extrinsic factors should be systematically considered.

## 4. Conclusions

In summary, the Postgraduate Research Flow Experience Scale developed in this study meets psychometric standards and is a valid instrument for assessing postgraduate research flow. The scale comprises 15 items across three dimensions: Clear Goals, Concentration, and Autotelic Experience. It employs a Likert 5-point scale (1 = “strongly disagree” to 5 = “strongly agree”), with higher scores indicating greater flow experiences in research.

As a key construct in positive psychology, flow exerts a pronounced enabling effect on postgraduate research. The scale establishes a three-dimensional model (Clear Goals, Concentration, and Autotelic Experience) of research flow experience. On the one hand, it offers a scientifically validated instrument for assessing and enhancing postgraduate research flow experience, thereby providing a new perspective on improving research efficacy. On the other hand, it furnishes theoretical guidance for educational management and individual development, enabling educators and administrators to design optimized research environments that facilitate flow states and, in turn, boost both research productivity and individual well-being.

This study has several limitations regarding its sample. First, the data were collected exclusively in mainland China, which may restrict the generalizability of the findings to other cultural or national contexts. Second, the sample size, while adequate for the primary analyses, constrains the statistical power for more complex modeling and subgroup comparisons. Future research would benefit from employing larger samples to validate the findings and enhance their external validity. Additionally, future studies could also integrate neuroscientific techniques (e.g., eye-tracking, fMRI) to further validate neural mechanisms underlying flow components. Furthermore, longitudinal research could also explore the long-term impact of flow experiences on research careers, and cross-cultural studies may verify the scale’s applicability across diverse academic contexts.

## Figures and Tables

**Figure 1 behavsci-15-01453-f001:**
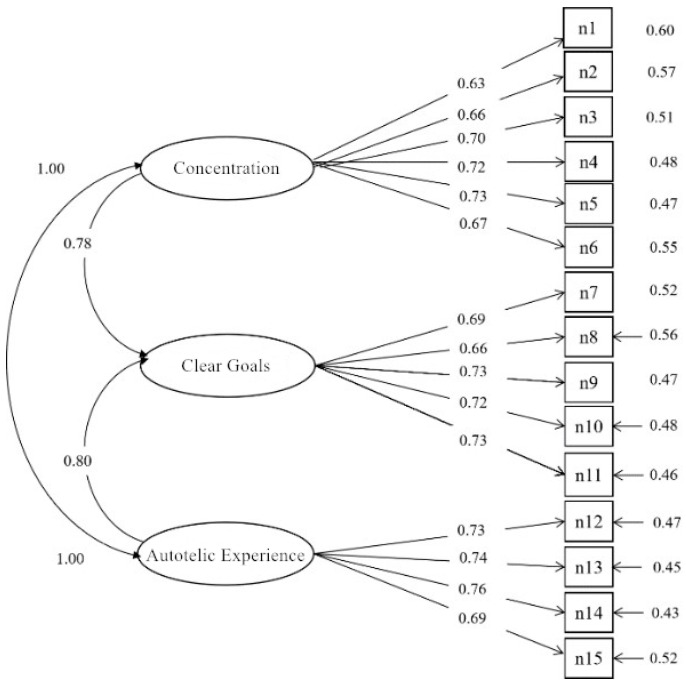
Structural Model of the Postgraduate Research Flow Experience Scale.

**Table 1 behavsci-15-01453-t001:** Exploratory Factor Analysis (EFA) of the Postgraduate Research Flow Experience Scale (N = 133).

Item No. (Factor Loading)							Eigenvalue	Percentage of Explained Variance
Factor 1	Q14(0.779)	Q12(0.766)	Q 11(0.759)	Q 26(0.750)	Q 22(0.748)	Q 13(0.664)	3.393	26.218%
Factor 2	Q 3(0.758)	Q 10(0.750)	Q 8(0.721)	Q 4(0.610)	Q 16(0.571)		3.114	46.976%
Factor 3	Q 24(0.796)	Q 18(0.749)	Q 5(0.695)	Q 19(0.638)			2.804	65.671%

**Table 2 behavsci-15-01453-t002:** Confirmatory Factor Analysis (CFA) of the Postgraduate Research Flow Experience Scale (N = 944).

	x^2^/df	RMSEA	CFI	NFI
Model	4.974	0.065	0.944	0.933

**Table 3 behavsci-15-01453-t003:** Correlations between the Postgraduate Research Flow Experience Scale and the criterion EduFlow Scale.

	Clear Goals (Flow Antecedents)	Concentration (Flow Experience)	Autotelic Experience (Flow Consequences)	Full Scale
EduFlow Scale	0.565 **	0.518 **	0.433 **	0.586 **
Cognitive Absorption	0.533 **	0.458 **	0.406 **	0.537 **
Transformation of Time	0.446 **	0.400 **	0.407 **	0.479 **
Loss of Self-consciousness	0.361 **	0.332 **	0.179 **	0.344 **
Autotelic Experience-well-being	0.474 **	0.472 **	0.419 **	0.526 **

** *p* ≤ 0.01.

**Table 4 behavsci-15-01453-t004:** Reliability Analysis of the Postgraduate Research Flow Experience Scale.

	Clear Goals (Flow Antecedents)	Concentration (Flow Experience)	Autotelic Experience (Flow Consequences)	Full Scale
Cronbach’s α coefficient	0.835	0.841	0.818	0.913
Guttman split-half reliability	0.786	0.814	0.800	0.826
McDonald’s omega	0.835	0.843	0.820	0.914

## Data Availability

The datasets generated during and/or analyzed during the current study are available from the corresponding author on reasonable request.

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
