# Peer review of "Postgraduate Research Flow Experience Scale: Development and Preliminary Validation"

_behavsci, 2025, doi:10.3390/bs15111453_

Round 1
Reviewer 1 Report
Comments and Suggestions for Authors
Dear Authors,
I have read your manuscript with much interest. I have found your research well conducted, accurately presented and adequately discussed.
I have only two minor comments for you:
- In page 4, line 174 you stated "Confirmatory factor analysis (CFA) yielded excellent model fit": it should be better to use good in place of excellent, as some of the indexes do not meet the acceptable thresholds (Chi-sq/df should not exceed 3) and CFI > .95 to be considered as excellent (Hu & Bentler, 1999).
- Please add some practical implications of the results of your study.
Author Response
Comments 1: In page 4, line 174 you stated "Confirmatory factor analysis (CFA) yielded excellent model fit": it should be better to use good in place of excellent, as some of the indexes do not meet the acceptable thresholds (Chi-sq/df should not exceed 3) and CFI > .95 to be considered as excellent (Hu & Bentler, 1999).
Response 1: Thank you for this insightful comment and for pointing out the need for greater precision in our terminology. We agree with your assessment. In the revised manuscript, we have changed the phrase "excellent model fit" to "good model fit" on Page 4, Line 174. We have also carefully reviewed and updated any similar descriptions elsewhere in the manuscript to ensure terminological consistency. We appreciate your guidance in enhancing the precision of our work.
Comments 2: Please add some practical implications of the results of your study.
Response 2: Thank you very much for suggesting that we elaborate on the practical implications of our findings. In response, we have significantly expanded the Discussion section (page 6, lines 256-272) to outline the practical applications of our study. The added content describes how the scale can inform personalized training interventions based on students' dimensional scores. For instance, students with low concentration or unclear goals can receive phased goal-setting training, while those scoring low on the "Autotelic Experience" dimension can benefit from visual progress trackers to navigate extended research projects. Furthermore, the scale's suitability for large-scale administration enables early intervention systems, where alerts can be triggered to supervisors or psychological services when scores fall below critical thresholds, helping to prevent research fatigue from escalating into more severe issues. Finally, the instrument provides quantifiable indicators for assessing mentoring quality and training outcomes at various administrative levels. We believe these additions substantially strengthen the practical relevance of our work.
Reviewer 2 Report
Comments and Suggestions for Authors
Thank you for allowing me to review this manuscript. The topic was succinctly described to understand the goal of the study. The methods were clearly addressed and the results were easy to follow. Sufficient support was given to the reliability and multiple types of validity. The conclusion tied the research to the results of this validation study. Perhaps the only issues that were apparent were of a formatting/copy editing nature; table and in-text citations did not appear as typical of the MDPI formatting.
Author Response
Comments 1: Thank you for allowing me to review this manuscript. The topic was succinctly described to understand the goal of the study. The methods were clearly addressed and the results were easy to follow. Sufficient support was given to the reliability and multiple types of validity. The conclusion tied the research to the results of this validation study. Perhaps the only issues that were apparent were of a formatting/copy editing nature; table and in-text citations did not appear as typical of the MDPI formatting.
Response 1: We sincerely appreciate you pointing out the issues regarding formatting and copy editing. We completely agree that adhering to the journal's style guide is crucial. In response to your comment, we have taken actions to ensure the manuscript fully complies with MDPI's formatting requirements.
Reviewer 3 Report
Comments and Suggestions for Authors
The aim of the study, as outlined in the introduction and abstract, is to develop and preliminarily validate a specific scale to measure the flow experience among postgraduate researchers during research activities. In particular, the study seeks to design an instrument that assesses the dimensions of “clear goals,” “concentration,” and “autotelic experience” within the context of academic work, with the purpose of providing a reliable and valid tool to evaluate this experience and, consequently, to support both educational processes and psychological assistance for postgraduate students.
The theoretical framework is based on Csikszentmihalyi’s flow theory, although its application to the specific context of postgraduate research could have been explored in greater depth. While the literature review does present existing instruments, it does not sufficiently incorporate recent theoretical approaches to measuring flow in diverse academic contexts, thereby limiting the theoretical justification for the creation of the new instrument. For instance, although motivation and intrinsic satisfaction are mentioned in relation to flow, no clear or detailed preliminary definition of these concepts is provided before introducing the autotelic experience. In this sense, the authors seem to assume these ideas as given, without offering a broader conceptual framework that could have enriched and better contextualized the study.
From a methodological standpoint, although the statistical analyses and model validation are sound, the explanation of the three dimensions —Clear Goals, Concentration, Autotelic Experience— lacks a broader theoretical discussion explicitly linking these aspects to the flow experience in academic research.
The results demonstrate good reliability; however, the near-exclusive reliance on statistical analyses limits the possibility of a deeper contextual interpretation. An insufficient conceptualization of the dimensions could, in turn, restrict the practical applicability of the instrument.
The discussion, while highlighting the main contributions of the study and acknowledging certain limitations, remains incomplete. Although emotional, motivational, and contextual factors not considered in the scale are briefly mentioned, there is no detailed examination of how external aspects—such as interruptions, technological distractions, or personal issues—may influence the flow experience in academic research. It would be worthwhile to revisit the theoretical aspects of the study in order to expand and deepen this section.
Finally, the study does not adequately address methodological limitations nor sufficiently discuss the potential lack of representativeness of the sample. A more thorough discussion of these aspects in future publications would enhance transparency and facilitate a more accurate interpretation of the findings, thereby strengthening the potential of the instrument as a tool for both psychological and academic support for postgraduate students.
Author Response
Comments 1: The aim of the study, as outlined in the introduction and abstract, is to develop and preliminarily validate a specific scale to measure the flow experience among postgraduate researchers during research activities. In particular, the study seeks to design an instrument that assesses the dimensions of ”clear goals”, ”concentration”, and ”autotelic experience” within the context of academic work, with the purpose of providing a reliable and valid tool to evaluate this experience and, consequently, to support both educational processes and psychological assistance for postgraduate students.
The theoretical framework is based on Csikszentmihalyi’s flow theory, although its application to the specific context of postgraduate research could have been explored in greater depth. While the literature review does present existing instruments, it does not sufficiently incorporate recent theoretical approaches to measuring flow in diverse academic contexts, thereby limiting the theoretical justification for the creation of the new instrument. For instance, although motivation and intrinsic satisfaction are mentioned in relation to flow, no clear or detailed preliminary definition of these concepts is provided before introducing the autotelic experience. In this sense, the authors seem to assume these ideas as given, without offering a broader conceptual framework that could have enriched and better contextualized the study.
Response 1:
Thank you for pointing this out. We agree with this comment. Therefore, we have expanded the research background by analyzing the applicability of established instruments (Page 2, Lines 79-86). While classic scales such as the FSS-2 have demonstrated value in specific contexts, they show limitations in capturing flow experiences within educational and scientific research settings. We have further emphasized the distinctive nature of scientific research activities in the revised manuscript (Page 2, Lines 92-93; Page 3, Lines 94-100). Research activities are characterized by extended timeframes and inherent challenges, exhibiting three core features: they demand autonomous metacognitive regulation through sustained self-monitoring and focused attention allocation; require management of nonlinear progression via strategic sub-goal anchoring to navigate phased fluctuations in knowledge innovation; and depend on sustained intrinsic motivation, where enduring interest drives continuous competence development. These distinctive characteristics necessitate the development of a context-sensitive measurement approach.
Comments 2: From a methodological standpoint, although the statistical analyses and model validation are sound, the explanation of the three dimensions-Clear Goals, Concentration, Autotelic Experience-lacks a broader theoretical discussion explicitly linking these aspects to the flow experience in academic research.
Response 2: We sincerely appreciate you pointing out the issues. We carefully analyzed the characteristics of flow in graduate research, which typically unfolds in multiple stages, such as literature review, protocol design, experiment execution, data collection, data analysis, paper writing, submission, and revision. Examining these activities requires attending to the situational variables at task initiation, the subjective experiences that occur during the task, and the behavioral and psychological outcomes that follow its completion. The three stages model of flow, with its “flow antecedents, flow experience, and flow consequences” framework, can therefore fully capture the dynamic process of flow and aligns well with the nature of graduate research. Building on this theoretical foundation and integrating our statistical results, we named the three factors of our questionnaire “Clear Goals”, ”Concentration”, and “Autotelic Experience” to link them to the flow experience in scientific research. We have added a description of this rationale in the Introduction section of the revised manuscript.
Comments 3: The results demonstrate good reliability; however, the near-exclusive reliance on statistical analyses limits the possibility of a deeper contextual interpretation. An insufficient conceptualization of the dimensions could, in turn, restrict the practical applicability of the instrument.
Response 3: We sincerely thank you for this valuable suggestion. We have revised the manuscript (Page 5, Lines 208-228; Page 6, Lines 239-255) to provide clearer conceptual explanations of the "Clear Goals," "Concentration," and "Autotelic Experience" dimensions within the research flow context. Each dimension is now explicitly illustrated with concrete daily examples from postgraduate research activities. Furthermore, we have expanded the discussion on the practical utility of the scale, strengthened the potential of the instrument as a tool for both psychological and academic support for postgraduate students.
Comments 4: The discussion, while highlighting the main contributions of the study and acknowledging certain limitations, remains incomplete. Although emotional, motivational, and contextual factors not considered in the scale are briefly mentioned, there is no detailed examination of how external aspects, such as interruptions, technological distractions, or personal issues, may influence the flow experience in academic research. It would be worthwhile to revisit the theoretical aspects of the study in order to expand and deepen this section.
Response 4: We sincerely thank you for this valuable suggestion. We have substantially expanded and refined the Discussion section (Page 6, Lines 256-282; Page 7, Lines 283-286). Specifically, we now provide an in-depth analysis of the three dimensions of the questionnaire, summarizing its core diagnostic functions and practical application value. Furthermore, we have added a dedicated subsection outlining the necessary conditions for its proper use. This includes a clear guideline stating that the interpretation of questionnaire results must integratively consider respondents' intrinsic factors (e.g., emotional state, motivation, personal issues) and external situational factors (e.g., research environment, technical problems, equipment failures) to ensure accurate and contextually grounded assessments.
Comments 5: Finally, the study does not adequately address methodological limitations nor sufficiently discuss the potential lack of representativeness of the sample. A more thorough discussion of these aspects in future publications would enhance transparency and facilitate a more accurate interpretation of the findings, thereby strengthening the potential of the instrument as a tool for both psychological and academic support for postgraduate students.
Response 5: We sincerely thank you for your insightful comments.In the revised manuscript, we have added a dedicated section in the conclusion part to explicitly address the methodological and sampling limitations (Page 7, Lines 300-313). We acknowledge that our sample was drawn exclusively from mainland China, and have now proposed concrete directions for future research-- including cross-national validation with expanded sample sizes and comparative cultural studies. Furthermore, we have significantly strengthened the discussion on the practical applications of the scale, highlighting its utility in monitoring research engagement, guiding supervisory support, and helping institutions provide a favorable external environment for postgraduate research activities.
Reviewer 4 Report
Comments and Suggestions for Authors
This study presents the development of the Postgraduate Research Flow Experience Scale and its preliminary validation by examination of its psychometric properties, including relaibility and validity. The topic of the flow is highly relevant in today’s educational context, including in higher education and postgraduate programs as flow is connected to intrinsic motivation involved in learning. The introduction provides a theoretical approach to the concept of flow, and a brief history on the research on flow, as well as arguments for studying it and developing a flow scale.
The methodology for elaborating the postgraduate research flow scale and validating it is minutely described by mentioning the item construction and selection process, paartcicipants involved, and data statistical analysis for the purpose of psychometric evaluations.
The conclusions rely on the statistical data analysis, and supplementary materials are available on the mdpi platform, as well as databases upon reasonable request.
The references are relevant and of recent date.
The study contributes to new knoweldge int he field by the elaboration and validation of the Postgraduate Research Flow Experience Scale.
Author Response
Comments 1: This study presents the development of the Postgraduate Research Flow Experience Scale and its preliminary validation by examination of its psychometric properties, including relaibility and validity. The topic of the flow is highly relevant in today’s educational context, including in higher education and postgraduate programs as flow is connected to intrinsic motivation involved in learning. The introduction provides a theoretical approach to the concept of flow, and a brief history on the research on flow, as well as arguments for studying it and developing a flow scale.
The methodology for elaborating the postgraduate research flow scale and validating it is minutely described by mentioning the item construction and selection process, paartcicipants involved, and data statistical analysis for the purpose of psychometric evaluations.
The conclusions rely on the statistical data analysis, and supplementary materials are available on the mdpi platform, as well as databases upon reasonable request.
The references are relevant and of recent date.
The study contributes to new knoweldge int he field by the elaboration and validation of the Postgraduate Research Flow Experience Scale.
Response 1: We sincerely thank the reviewer for their positive feedback on our work and their meticulous evaluation of the manuscript. We are greatly encouraged by the recognition of our research efforts.V
Round 2
Reviewer 3 Report
Comments and Suggestions for Authors
The authors have thoroughly incorporated the suggested revisions, significantly enhancing the theoretical and contextual grounding of the study. First, they expanded the background by critically examining the applicability and limitations of existing flow measurement scales, thereby establishing the need for a research-specific instrument. They also provided a detailed explanation of the stages of flow in postgraduate research, clearly aligning each stage with the questionnaire’s dimensions: Clear Goals, Concentration, and Autotelic Experience. In addition, concrete examples from daily academic activities were added to illustrate how these dimensions emerge throughout the research process.
The Discussion section has been strengthened to highlight the diagnostic and practical value of the instrument, along with the necessary conditions for its correct application. Finally, the authors included a new section addressing methodological and sampling limitations and proposed clear directions for future research, emphasizing cross-cultural validation and the instrument’s potential to inform academic support strategies and institutional policies. Therefore, I believe that it can be published.